# Correcting the Non-Linear Response of Silicon Photomultipliers

**DOI:** 10.3390/s24051671

**Published:** 2024-03-05

**Authors:** Lukas Brinkmann, Erika Garutti, Stephan Martens, Joern Schwandt

**Affiliations:** Institute for Experimental Physics, University of Hamburg, Luruper Chaussee 149, 22761 Hamburg, Germanyjoern.schwandt@desy.de (J.S.)

**Keywords:** SiPM, non-linearity, response function

## Abstract

The finite number of pixels in a silicon photomultiplier (SiPM) limits its dynamic range to light pulses up to typically 80% of the total number of pixels in a device. Correcting the non-linear response is essential to extend the SiPM’s dynamic range. One challenge in determining the non-linear response correction is providing a reference linear light source. Instead, the single-step method used to calibrate PMTs is applied, based on the difference in responses to two light sources. With this method, the response of an HPK SiPM (S14160-1315PS) is corrected to linearity within 5% while extending the linear dynamic range by a factor larger than ten. The study shows that the response function does not vary by more than 5% for a variation in the operating voltage between 2 and 5 V overvoltage in the gate length between 20 and 100 ns and for a time delay between the primary and secondary light of up to 40 ns.

## 1. Introduction

Silicon photomultipliers (SiPMs) are solid-state photo-detectors consisting of a matrix of single-photon avalanche diodes (SPADs) called pixels or cells. A SiPM operates in Geiger mode and is able to detect and count single photons with good time resolution. Among the advantages of SiPMs are the low operating voltage, compactness, and insensitivity to magnetic fields. Those qualities make SiPMs the best choice for high-granularity detectors for applications in high-energy physics and medical imaging.

Due to the Geiger mode operation, each pixel is a binary counter and delivers the same charge independent of the number of photons instantaneously impinging on it. The SiPM signal is given by the parallel sum of all pixel charges. After a photon is detected, the Geiger avalanche is quenched by dropping the pixel voltage below breakdown via a quenching resistor. The pixel recharges to its operational status after a given time. Photons that arrive on the discharging pixel before this characteristic time may not be detected or be detected with a smaller charge.

In this study, we develop a setup to measure the response of an SiPM as a function of the light intensity, and under various conditions such as different operation voltages and temperatures, and a method to apply this as a correction of the SiPM non-linearity. The measured SiPM signal as a function of light intensity suffices for the computation of the correction function. The correction is independent of the precise knowledge and linearity of the light source. It is based on the single-step method introduced by Gatti and Piva in 1953 and described by Wright in [1] which uses two superimposed light sources. While the superposition happens in space, the two light sources can be shifted in time, allowing us to study the effect of the light temporal distribution on the response function. The measurement of saturation effects in SiPMs with a single light source and the phenomenological parameterization of these have been discussed in detail in the literature, e.g., in [2,3]. In [4], SiPMs with 100, 400, 1600, and 2668 pixels with sub-nanosecond light pulses of different intensities were investigated; in [5], SiPMs were exposed to light pulses of different intensities and pulse lengths; and, in [6], saturation effects of an SiPM using UV light are investigated. More measurements presented in the literature are summarized in the review [7,8]. All these methods require calibration of the reference light source, which is avoided with the single-stage method.

The paper is organized as follows: Section 2 introduces the single-step method, the novel setup developed for the SiPM non-linearity measurements, and the key parameters of the SiPM used for the study. Section 3 discusses how measurements are taken and presents an exemplary data set. Finally, Section 4 presents the correction method and the results of the response linearization. In Section 5, the conclusions of the study are given.

## 2. Materials and Methods

According to the single-step method described in [1], the non-linearity of a photo-detector is determined by measuring the changing charge of small fixed light pulse dφ when added onto the primary variable base pulse $\mu_{L}$. For this method, two sets of measurements are recorded: the SiPM response to the variable pulse $\mu_{L}$ and the response to the sum of the variable pulse and base fixed light pulse, $\mu_{L}+d\mu =\mu_{\mathrm{LL}}$. Note that $\mu_{L}+d\mu$ differs from $\mu_{L}+d\varphi$ as one measures the “effective” added amplitude of the small fixed light pulse. The difference between dμ and dφ gives the measurement of non-linearity. For a linear device, this should always be equal to the charge dφ. As the non-linearity increases, the difference tends towards zero, at which point, none of the added fixed light pulse is detected.

The dedicated setup sketched in Figure 1 was built to realize the single-step method.

The essential aspect of the setup is to provide uniform light distributed over the entire SiPM surface. The light consists of two sources separately tunable in intensity and time. The variable-intensity base pulse is generated with a Laser. The optical Laser head is the Picosecond Injection LASER (PiLas) PiL044SM from Advanced Laser Diode Systems, Berlin, Germany [9] with a pulse length of t=50 ps and a wavelength of $\lambda_{\mathrm{Laser}}=451$ nm. The Laser control unit used is the EI61000D from Advanced Laser Diode Systems. It triggers on the rising edge of the pulse from the pulse generator. The internal TUNE value controlling the intensity of the Laser light is set to 50% to ensure the use of the Laser in its optimal working regime. The Laser is coupled into a multimode fiber, indicated as F1 (Thorlabs BFY400HF2, L=2 m with FT400UMT fiber; diameter, $D_{\mathrm{Core}}=105\;\mu m$; $D_{\mathrm{Cladding}}=125\;\mu m$; and numerical aperture, NA=0.22). To change the intensity of the Laser, different neutral-density filters are used. The calibration of the optical density is documented in [10], but, as later shown, it is not necessary for the correction. From the neutral-density filters, the light is coupled into the multimode fiber F3 (Thorlabs FG365UEC, L=20 m, $D_{\mathrm{Core}}=365\;\mu m$, $D_{\mathrm{Cladding}}=400\;\mu m$, NA=0.22).

The constant-intensity pulse dφ is produced by an LED. The LED control unit is the Sepia 2 PDL 828 from PicoQuant, Berlin, Germany [11], and the LED head is the Sub-Nanosecond Pulsed LED PLS 450. The LED is also triggered on the rising edge of the trigger, and the intensity of the LED light can be set in the driver. The length of an LED light pulse is t=980 ps with a wavelength of $\lambda_{\mathrm{Led}}=460\;nm$ and a spectral width of 40 nm. The LED is coupled into a 50/50 multimode fiber coupler F2 (Thorlabs FG365UEC, L=1.6 m, $D_{\mathrm{Core}}=400\;\mu m$, $D_{\mathrm{Cladding}}=425\;\mu m$, NA=0.39). It is used to split 50% of the light from the input port into the signal port and 50% into the tap port. The ports allow the propagation of different modes. The LED stability is quantified to have a maximum variation of 3% over several hours. This value is used as uncertainty on the constant LED light in the later analysis.

The LED and the Laser are coupled into one single bifurcated fiber F4 (Thorlabs BFY400HF2, L=2 m with FT400UMT fiber; $D_{\mathrm{Core}}=400\;\mu m$, $D_{\mathrm{Cladding}}=425\;\mu m$, NA=0.39) which enters the climate chamber and illuminates the SiPM. The fiber has two split ends that serve as the input for the Laser and LED light, respectively. The distance between the cores in the bifurcated fiber is separated only by 55 μm; a sketch from the producer is provided in Figure 2. The distance from the fiber end to the SiPM front face is tuned to have the SiPM within one standard deviation from both light cones. This is achieved for distances of around d=7.5 mm. Since light spots exhibit Gaussian intensity profiles, the maximum decrease in light over the SiPM area at the chosen distance is 60%. The uniformity of the light could be enhanced by further increasing the distance at the price of a decreased maximum intensity.

A Stanford Research System Model DG645 [13] delay generator serves as the trigger for the Laser, LED, and oscilloscope. With the delay generator, the timing of the Laser and LED can be adjusted separately. A delay time of zero ns indicates that the rising edges of the SiPM pulses triggered by Laser light only and LED only match. The trigger pulse has a length of 50 ns and a height of 500 mV, and it is set to a rate of 1 kHz.

The device under investigation is the SiPM S14160-1315PS from Hamamatsu Photonics K.K. [14] (HPK). The key properties of this SiPM are reported in Table 1. The parameters in various blocks are taken from (i) the producer’s specification, (ii) laboratory measurements, (iii) a double-exponential fit to the average waveform, and (iv) a fit performed with PeakOTron-1.0.0 [15,16] software to the single-photoelectron peak spectrum at reference operating conditions (20 °C, $V_{\mathrm{OV}}=3.94\;V$).

## 3. Measurements

To measure the SiPM non-linearity curve, two sets of measurements are recorded: the SiPM response to the variable Laser pulse $\mu_{L}$ and the SiPM response to the Laser plus fixed LED light, $\mu_{\mathrm{LL}}$. The measured LED light intensity, in number of photoelectrons (npe), is fixed to about 5% of the total number of pixels, dφ∼ 300 npe. A Rohde & Schwarz RTO2044 oscilloscope [18] is used for the data acquisition. For each light amplitude setting, 50,000 single waveforms are stored and later processed by a Python-based analysis code. The waveforms at fixed light intensity are integrated over a gate with a tunable length, and the moments of the distributions for each settings are calculated. If not differently specified, a gate of 100 ns is used, which starts 5 ns before the SiPM pulse.

The first three moments and their errors are calculated for each charge distribution Q[n] with a fixed light setting. For the second and third, the central moments are used.
(1)$$\begin{array}{cc} m_{i} & =\sum_{n=1}^{N}{\left(Q\left[n\right]-\mu_{1}\right)}^{i},\;\;\;\;\;\;\;\;\;i>1 \end{array}$$
(2)$$\begin{array}{cc} \mu =\mu_{1} & =\frac{1}{N}\sum_{n=1}^{N}\left(Q\left[n\right]\right);\;\;\;\;\;\;\;\;\;\;\;\;\;\;\delta \mu_{1}=\sqrt{\frac{\mu_{2}}{N}} \end{array}$$
(3)$$\begin{array}{cc} \mathrm{RMS}=\sqrt{\mu_{2}} & =\sqrt{\frac{1}{N-1}m_{2}};\;\;\;\;\;\;\;\;\;\;\;\;\;\;\;\;\delta \mu_{2}=\sqrt{\frac{\mu_{2}}{2(N-1)}} \end{array}$$
(4)$$\begin{array}{cc} \mathrm{sk}=\mu_{3} & =\frac{N\sqrt{N-1}}{N-2}\frac{m_{3}}{m_{2}^{3/2}};\;\;\;\;\;\;\;\;\delta \mu_{3}=\sqrt{\frac{6N(N-1)}{\left(N-2)(N+1)(N+3\right)}} \end{array}$$

For the first and second moments, the respective dark moments, the moments of the pedestal waveform in the absence of a light pulse are subtracted.

Figure 3 presents the mean values of $\mu_{L}$ and $\mu_{\mathrm{LL}}$ versus the linear Laser light intensity calibrated using neutral-density filters.

The Laser light intensity is given in a percentage to its maximum value. In the following plots, this axis is rescaled, setting the lowest light intensity measurement as μ = 80 npe, equal to the number of seed photons, $N_{\mathrm{seed}}$. This calibration assumes a linear response at low light intensity and negligible correlated noise. Note that knowledge of this linear axis is not required in the method discussed in this paper, but this representation of the SiPM response is used for illustration purposes.

The *y*-axis is converted to the unitless number of photoelectrons, or npe, by dividing the measured charge by the charge of the detection of one single photon as follows:(5)$$\mu \left[npe\right]=\frac{\mu [V\;s]}{R_{L}\;\cdot \;e\;\cdot \;G}\;,$$
where $R_{L}$ is the load resistance at the oscilloscope, 50 Ω, and *e* is the electron charge and the gain of the SiPM *G* at the temperature, voltage, and gate length of the measurements.

For low light intensity, the measured charges from the Laser+LED and Laser illumination differ by the expected value of dφ∼ 300 npe. The difference decreases as the Laser intensity increases. The two inserts show the very-low- and very-high-light-intensity regions zoomed in, in linear scale, for better visibility. Both the Laser and Laser+LED measurements are linear in this range. The zoomed-in image of the high-intensity range shows that the SiPM response exceeds the total number of physical pixels of the SiPM, $N_{\mathrm{pix}}=7296$. Furthermore, it can be seen that full saturation does not occur for this SiPM even for a light pulse duration shorter than 1 ns. This contradicts expectations and deserves a separate investigation.

Figure 4 shows the RMS and skewness of the charge spectra as a function of light intensity calibrated to the average number of seed photons. The skewness is consistent with zero for light intensity $N_{\mathrm{seed}}>1000$, indicating that, for a large number of detected photons, the Poisson distribution tends to be Gaussian. The RMS increases with the light intensity, reaching a maximum shortly before the number of seed photons matches the total number of pixels. It then decreases to a minimum similar to the value for low light intensity. This effect is ascribed to the strong non-linearity of the response, which reduces the spread of the distribution in a non-Poissonian way. Indeed, the skewness for these light intensities becomes slightly negative. The second growth of the RMS after the minimum for light intensities exceeds the total number of pixels by about a factor of 8 is not yet understood.

The RMS of the Laser+LED light for low light intensity is broader than the RMS of the Laser only for the same number of seed photons. The increase in RMS might be due to the difference in pulse widths. While the Laser pulse width is 50 ps, the LED light has a pulse width of 980 ps. This can be better investigated by delaying the LED light relative to the Laser and studying the effect on the RMS.

In Figure 5, late-arriving LED light increases both the mean and the RMS of the collected charge distributions compared to synchronous light sources. The largest deviation occurs when the number of seed photons is roughly equal to the number of pixels. The differences in the mean are below 5% up to a time delay of 40 ns for roughly 300 photons from the LED. The RMS increases by a factor of up to 3, most likely due to the large fluctuations in the pixel recovery status at the time of the second photon detection in a single pixel.

This measurement simplifies the real experimental conditions where, typically, photons are distributed with a continuous time spectrum and may have a tail of up to several tens of ns. The study only indicates the necessity to calibrate the SiPM response function with the exact light distribution expected in the final application.

## 4. Non-Linearity Correction

This paper aims to extract a correction of the SiPM non-linear response, which is independent of the linearity of the reference light signal or the precision of its calibration. Differently formulated, the correction should only depend on the measured SiPM charge. The measured LED intensity of the single step is denoted as dφ. For a detector with linear response, it is expected that, at every single step, dφ is equal to dμ, the difference between the mean charges with illumination by Laser+LED light ($\mu_{\mathrm{LL}}$) and Laser light only ($\mu_{L}$). The slope of the response function is therefore as follows:(6)$$S\left(\mu \right)=\frac{d\mu }{d\varphi }=\frac{\left(\mu_{\mathrm{LL}}-\mu_{L}\right)}{d\varphi }.$$

Notably, Equation (Equation 6) only depends on measured quantities.

The reference non-linearity curve for the SiPM under study operated at *T* = 20 °C, $V_{\mathrm{over}}=3.94$ V, and for dt=0.0 ns using an integration gate of 100 ns is presented in Figure 6. The *x*-axis is the mean value of $\mu_{\mathrm{LL}}$ and $\mu_{L}$. The data are smoothed by a linear spline function. This function is used in the rest of the paper to correct data from the same SiPM operated in different conditions.

The corrected response $\varphi =\mu_{\mathrm{lin}}$ for measured charge value μ is calculated, integrating the inverse of the reference non-linearity curve S(μ) from zero up to μ as follows:(7)$$\begin{array}{cc} \varphi & =\int_{0}^{\mu }\frac{1}{S(\mu )}d\mu . \end{array}$$

The integration over dμ is implemented as the cumulative sum of the function times the step size dμ. Thanks to the use of the spline, the step size can be chosen arbitrarily with respect to the reference curve measurement steps, but the precision of the correction will depend on the accuracy of the reference curve.

The corrected SiPM response φ obtained using Equation (Equation 7) on the $\mu_{L}$ data presented in Figure 3 is shown in Figure 7. Two methods to apply the correction are shown: correction of the mean of a set of data at equal light intensity and the correction event by event of each measurement. In both cases, the linearity of the corrected data is improved. The right plot shows that, after event-by-event correction, the RMS of the charge response scales as expected with $N_{\mathrm{seed}}^{0.5}$ up to a point where the data are slightly overcorrected.

In the left plot of Figure 8, the response linearity is plotted by subtracting a slope of one from the data of Figure 7. The uncorrected data diverge from linearity by more than ±5% for $N_{\mathrm{seed}}\geq 1000$∼$15\%\;\cdot \;N_{\mathrm{pix}}$. The mean corrected data stay within ±5% of linearity up to $N_{\mathrm{seed}}\geq$ 45,000∼$6\;\cdot \;N_{\mathrm{pix}}$, excluding the single outlier at $N_{\mathrm{seed}}$∼$N_{\mathrm{pix}}=7296$. If the linear dynamic range is defined as the min-to-max range in which the SiPM response is linear within ±5%, the correction increases the linear dynamic range by a factor of ∼45.

The right plot of Figure 8 demonstrates the result of applying the reference calibration function from Figure 6 to data integrated for different gate lengths between 20 ns and the reference 100 ns. The response can also be linearized to ±5% up to a signal equal to $N_{\mathrm{pix}}$ if the integration gate of the data is significantly shorter than that used to determine the calibration curve.

Figure 9 presents two further systematic studies where the reference calibration function is applied to data collected with different overvoltages or with a different delay between the two light sources. Also, in this case, up to a signal equal to $N_{\mathrm{pix}}$, the response is linear within ±5%.

## 5. Conclusions

The single-step method to correct the non-linear response of a photo-detector is applied to a Hamamatsu SiPM (S14160-1315PS) with 15 μm pixel pitch and 7296 pixels operated at different voltages.

The main strength of the method presented is that the calibration function can be obtained in a laboratory setup and is independent of the knowledge and linearity of the reference light source.

For the measurements, a novel setup was developed and characterized to provide homogeneously distributed and overlapping light from two sources, an LED and a Laser, which can be individually tuned in amplitude and arrival time.

With this method, it is possible to correct the non-linear response of this specific SiPM and extend the linear dynamic range by a factor larger than ten.

The measurements show that the correction function does not depend on the operating voltage in the range between 2 and 5 V overvoltage. This makes it possible to measure the response function of an SiPM once and still correct it if the operating voltage is to change. Also, the dependence on the integration gate is minor, within 20–100 ns, for the specific signal shape of this SiPM. Most of the studies were performed by correcting the mean of light distributions at fixed light intensity. The event-by-event correction of each measured charge is also demonstrated to work. More detailed systematic studies on applying this correction event by event, including temperature, noise, radiation, and SiPM type dependencies, will be carried out in future studies.

To apply this method in practical applications it will be important to create a test stand where the SiPM is calibrated in conditions as close as possible to the those of the final experiments. Temperature, readout electronics, cooling, and light illumination should be close to those expected during operation. However, as is illustrated in this study, the correction appears to be stable against small variations in the operation parameters. More systematic studies will be needed to quantify this statement.

## Figures and Tables

**Figure 1 sensors-24-01671-f001:**
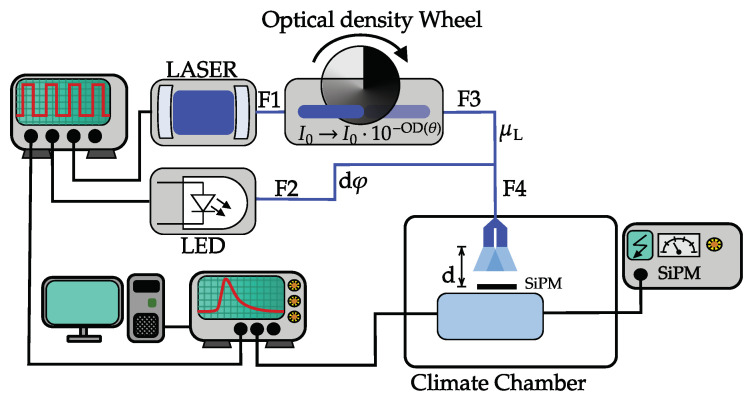
Non-linear response measurement setup schematics.

**Figure 2 sensors-24-01671-f002:**
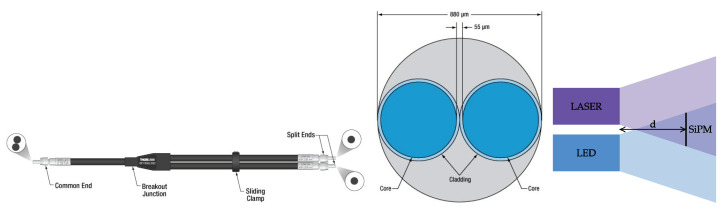
(**Left**) Sketch of the bifurcated fiber. The inputs from the split end (right end) are combined into one fiber at the common end (left end) [12]. (**Center**) Cross-section of the bifurcated fiber common end. (**Right**) Sketch of the ideal position of the SiPM in the area where both Laser and LED light overlap.

**Figure 3 sensors-24-01671-f003:**
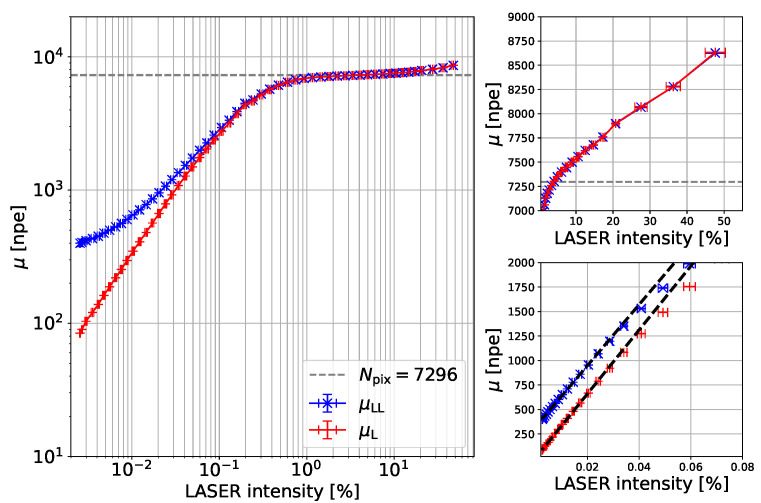
Mean values of Laser only (red) and Laser plus LED light (blue) versus Laser intensity. The two zoomed-in inserts are the high- and low-intensity regions at linear scale, respectively. The measurements are taken at 20 °C with zero time delay between LED and Laser light. The SiPM is operated at an overvoltage $V_{\mathrm{OV}}=3.94\;V$.

**Figure 4 sensors-24-01671-f004:**
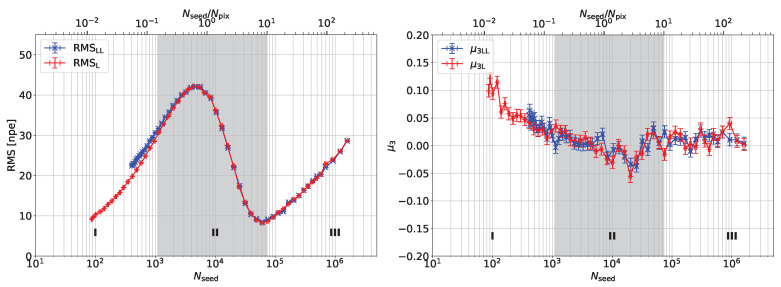
RMS values (**left**) and skewness (**right**) of Laser only (red) and Laser plus LED light (blue) versus the mean number of seed photons between Laser and Laser+LED for each measurement step. The Laser+LED data set is shifted on the *x*-axis by the value of dφ compared to the Laser one. The measurement conditions are the same as in Figure 3. Three regions are marked for discussion: (I) linear response within ±5%, (II) non-linear response, which can be recovered with the method discussed in this paper, (III) non-linear response which cannot be recovered.

**Figure 5 sensors-24-01671-f005:**
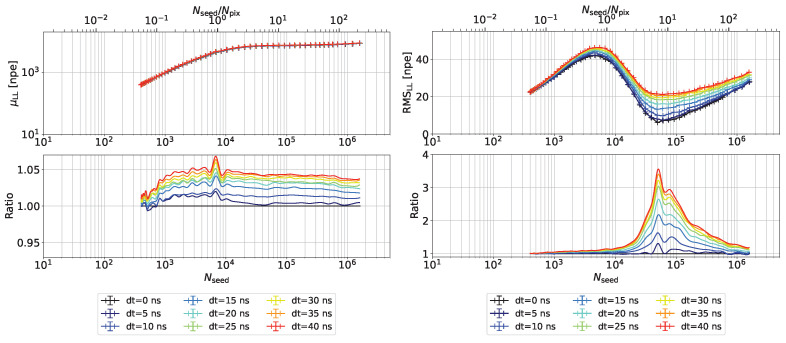
Mean (**left**) and RMS (**right**) of the charge spectrum for Laser+LED light for various delays between the light sources.

**Figure 6 sensors-24-01671-f006:**
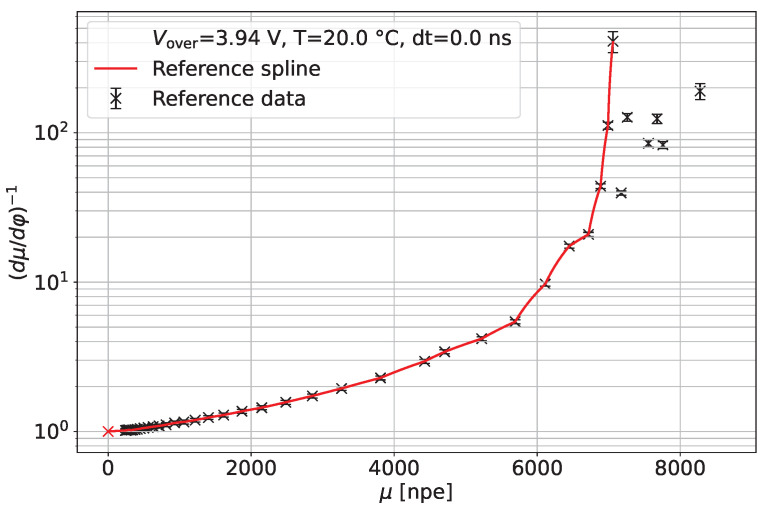
Correction function according to Equation (Equation 6). The measurements are taken at 20 °C and with zero time delay between LED and Laser light. The SiPM is operated at an overvoltage $V_{\mathrm{OV}}=3.94\;V$.

**Figure 7 sensors-24-01671-f007:**
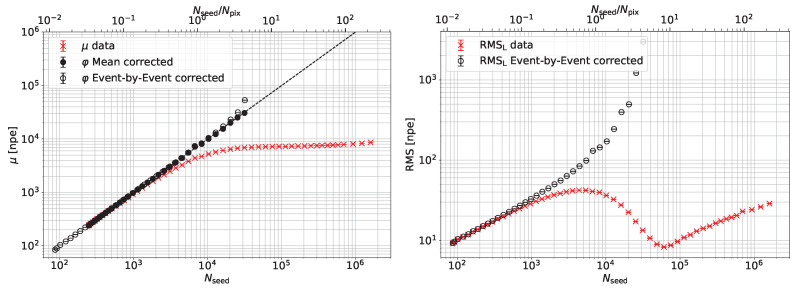
Mean value (**left**) and RMS (**right**) of the corrected SiPM response (black) compared to the uncorrected data (red). Open circles indicate the correction applied event by event, while, for the closed circle, the correction is applied to the charge mean for each light intensity.

**Figure 8 sensors-24-01671-f008:**
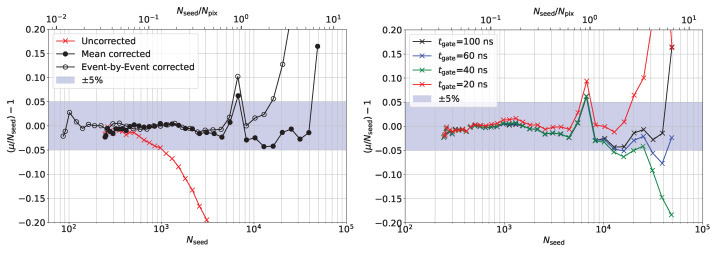
(**Left**) The normalized difference between data and expected linear response from Figure 7. (**Right**) Correction function applied to data integrated with different gate lengths.

**Figure 9 sensors-24-01671-f009:**
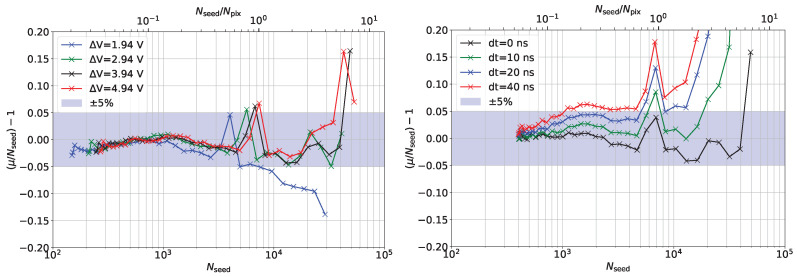
Correction function applied to overvoltage scan (**left**) and time delay scan (**right**).

**Table 1 sensors-24-01671-t001:** Key parameters of the device investigated in this study: the SiPM S14160-1315PS from Hamamatsu Photonics K.K. [14].

Block	Parameter	Symbol	Value
	Effective photosensitive area	-	1.3×1.3 mm^2^
	Pixel pitch	-	15 μm
i	Peak sensitivity wavelength	$\lambda_{p}$	460 nm
	Photon detection efficiency at $\lambda_{p}$	PDE	32%
	Number of pixels	$N_{\mathrm{pixel}}$	7296
	Breakdown voltage	$V_{\mathrm{br}}$	(37.27 ± 0.01) V
ii	Overvoltage	$V_{\mathrm{OV}}$	2 V to 5 V
	Load resistor	$R_{L}$	50 Ω
	Quench resistor	$R_{Q}$	780 kΩ [17]
	Fast time component	$\tau_{f}$	(0.90 ± 0.05) ns
iii	Slow time component	$\tau_{s}$	(14.4 ± 0.1) ns
	Fraction of fast component charge	$r_{f}$	0.035
	Dark count rate	DCR	(3.0 ± 0.2) MHz
iv	Crosstalk probability	λ	0.02
	Afterpulse probability	$p_{\mathrm{Ap}}$	0.02
	Time constant for afterpulse	$\tau_{p_{\mathrm{Ap}}}$	8.05 ns

## Data Availability

Data will be made available on request.

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
