# Peer review of "Correcting the Non-Linear Response of Silicon Photomultipliers"

_sensors, 2024, doi:10.3390/s24051671_

Round 1

Reviewer 1 Report

Comments and Suggestions for Authors

In their paper “Correcting the non-linear response of Silicon Photomultipliers,” the authors present a method for calibrating SiPMs akin to that of the method used for PMTs. This method could prove useful for improving the usability and performance of SiPMs with less calibration than is required by other methods. I recommend the publication of this manuscript following the resolution of the a few comments.

1.     On page 2, please note if the fibers are single mode or multimode and the fiber core diameters in addition to the cladding diameters. Please also note the spectral width of the LED.

2.     Why was a bifurcated fiber chosen rather than a WDM? Were collimating optics considered on the output of the fiber?

3.     Please add labels to Figure 1 indicating the light pulses dφ, dµ, µL, etc. where applicable

4.     Can the authors further explain how they accurately determined Nseed? It appears linearity is assumed at low light intensity in order to determine this value.

5.     How is the ND filter wheel characterized and correctly positioned? Please elaborate further on why this calibration is not critical as claimed in line 68.

6.     Can the authors further explain why the linearity seems to recurringly suffer near Nseed = Npix? Is this due to the limited spline fit near high values of µ as seen in Fig. 6? Can the authors explain the data observed in this region of Fig. 6?

7.     Can the authors comment on how the described linearity correction method compares to other methods? Is greater or similar performance achieved using this method?

Comments on the Quality of English Language

There are a few minor typos (line 85, line 137)

Author Response

Reviewer 1)

  1. On page 2, please note if the fibers are single mode or multimode and the fiber core diameters in addition to the cladding diameters. Please also note the spectral width of the LED.

We collected the following information and included them in the paper as far as appropriate.

F1: Laser -> filter-wheel collimator IN
Thorlabs FG105UCA / 10m

https://www.thorlabs.com/newgrouppage9.cfm?objectgroup_id=6838&pn=FG105UCA#6842

Ø105 µm Core Glass Clad Silica Multimode Optical Fiber, 0.22 NA
High OH -> 250-1200 nm

Core: Pure Silica, Ø 105 µm +1/-3 µm
Cladding: Fluorine-Doped Silica, Ø 125 µm +1/-2 µm

Connector 1: FC/APC (Laser head end)
Connector 2: FC/PC (Collimator end)

F2: LED     -> Bifurcated Fiber Bundle Arm 1 (with reference diode option)
Thorlabs TT400R5F1B with FT400EMT fiber

https://www.thorlabs.com/newgrouppage9.cfm?objectgroup_id=13747&pn=TT400R5F1B#13748

1x2 Fiber Optic Coupler
TECS-Clad Multimode Optical Fiber, Step Index, 0.39 NA
Low OH -> 400-2200 nm

Core: Ø 400 µm +/-8 µm
Cladding: Ø 425 µm +/-10 µm

Connector 1                        : FC/PC 2.0 mm (Input)
Connector 2a   : FC/PC 2.0 mm (Signal Output)
Connector 2b  : FC/PC 2.0 mm (Tap Output)

F3: 2. filter-wheel collimator OUT -> Bifurcated Fiber Bundle Arm 2
Thorlabs FG365UEC / 20m

https://www.thorlabs.com/newgrouppage9.cfm?objectgroup_id=6839&pn=FG365UEC#6881

Ø365 µm Core TECS-Coated Multimode Optical Fiber, 0.22 NA
High OH -> 250-1200 nm

Core: Pure Silica, Ø 365 µm +/-14 µm
Cladding: Fluorine-Doped Silica, Ø 400 µm +/-8 µm

Connector 1: FC/PC (Collimator end)
Connector 2: FC/PC (Bifurcated Fiber Bundle end)

F4: Bifurcated Fiber Bundle
Thorlabs BFY400HF2 / 2m with FT400UMT fiber

https://www.thorlabs.com/newgrouppage9.cfm?objectgroup_id=7510&pn=BFY400HF2#7511

TECS-Clad Multimode Optical Fiber, Step Index, 0.39 NA

High OH -> 300-1200 nm

Core: Ø 400 µm +/-8 µm
Cladding: Ø 425 µm +/-10 µm

Connector 1                        : FC/PC 2.0 mm Narrow Key (Common End)
Connector 2a   : FC/PC 2.0 mm Narrow Key (Split End)
Connector 2b  : FC/PC 2.0 mm Narrow Key (Split End)

PLS 450:

wavelength:                        460 +/- 10 nm                      (from specs sheet)
spectral width:                
40 nm                                       (from specs sheet)
pulse width:                        800 ps

  1. Why was a bifurcated fiber chosen rather than a WDM?

Setting up the experiment, we started with fiber F2 (1x2 Fiber Optic Coupler) to combine the two beams (the fiber type is the same as for the Bifurcated Fiber Bundle F4). Due to the high power losses for using it as a combiner and not as a splitter, we couldn’t use it for our saturation application.

For combining light of different wavelengths, Thorlabs also offers a line of single mode wavelength division multiplexers (WDMs), where each arm has got a designated wavelength range (e.g. one arm for blue/green light, the other one for red light). Combining two blue sources would lead to high insertion losses in one arm of the WDM.

So we have chosen a Multi-Mode Bifurcated Fiber Bundle to overcome the limited wavelength ranges of fused fiber couplers.

No change in the text

Were collimating optics considered on the output of the fiber?

By varying the distance of the FC/PC connector to our SiPM / CCD sensor, we could try out and investigate different patterns of the overlapping beams from laser/LED. Different sizes of MPPCs can be taken into account with this method.

Using an additional adjustable collimator could help to get a larger clearance for handling the sample under the beam output for the highest intensities.

Thank you for the suggestion. No change in the text.

  1. Please add labels to Figure 1 indicating the light pulses dφ, dµ, µL, etc. where applicable

Thank you for the suggestion. dφ, µL are now labelled in Fig. 1

  1. Can the authors further explain how they accurately determined Nseed? It appears linearity is assumed at low light intensity in order to determine this value.

The original sentence

“The laser light intensity is an arbitrary scale, which can be calibrated to the number of seed photons, $N_\mathrm{seed}$ using the lowest measurement light intensity assuming linear response and negligible correlated noise.”

has been modified to:

“The laser light intensity is given in percentage to its maximum value. In the following plots this axis is rescaled setting the lowest light intensity measurement $\mu$ = 80 npe equal to the number of seed photons, $N_\mathrm{seed}$. This calibration assumes linear response at low light intensity and negligible correlated noise.”

  1. How is the ND filter wheel characterized and correctly positioned? Please elaborate further on why this calibration is not critical as claimed in line 68.

The correction method introduced in this paper is based on the SiPM response only. The correction function is taken from Figure 6, which does not depend on source light intensity and it is applied to SiPM measurements with no knowledge of the source light intensity. The linear axis obtained from the density filter calibration is used for plotting purposes only, as the SiPM users are accustomed to plotting SiPM response vs linear light intensity. We therefore deem it unnecessary to include further details on the calibration of this axis.

No change to the text

  1. Can the authors further explain why the linearity seems to recurringly suffer near Nseed = Npix? Is this due to the limited spline fit near high values of µ as seen in Fig. 6? Can the authors explain the data observed in this region of Fig. 6?

Thank you for this question. Indeed we were puzzled ourselves and repeated the measurements many times and in different conditions. All curves present the same discontinuity near Nseed = Npix. When removing this one single point the spline performs better, but we could not find any experimental reason to reject the measurement. Data with other SiPMs will be taken soon to further investigate this point. For the moment we cannot explain, but we do not consider this single outlier a reason not to publish the method.

No change to the text

  1. Can the authors comment on how the described linearity correction method compares to other methods? Is greater or similar performance achieved using this method?

This is a very good question that would require a very complex comparative study. Some comments.

  • The method developed in this paper is the only one (to our knowledge) that does not require knowledge of the calibration and linearity of the light source. In this sense, it is already much simpler to apply than what is done in the literature.
  • The methods applied in the literature are based on fitting the response of the SiPM (assuming a linear reference source) with some function. Inverting the function and applying it to correct the response. The precision of this method depends on the accuracy of the calibration and the accuracy of the fit model chosen.
  • Many fit functions are presented in the literature, from the most simple single exponential to much more complex functions. Usually, they are applied to one type of SiPM operated in one specific condition. Seldom operation conditions are changed to test the validity of the model. Also, the applicability of models to various SiPMs are lacking (as in our study so far)

We do not feel ready to write such a statement in the paper but would prefer to deliver first a significant systematic study using our method before expressing such a comparative statement. For now, simplicity and model independence is the key aspect of our model. The 5% linearity obtained is more than adequate for the SiPM calibration, and comparable with if not better than existing literature results.  

No change to the text

Extra corrections.

While reading the paper we have additionally spotted a few more points that needed correction. We excuse ourselves to the reviewers for not spotting this earlier and hope thy will agree with these changes.

  • The definition of dynamic range was not provided, and given the non-linear behaviour it may be misleading. We introduced the concept of “linear dynamic range” with the following definition, provided now in the discussion of Fig. 8. “If the linear dynamic range is defined as the min-to-max range in which the SiPM response is linear within $\pm5\%$, the correction increases the linear dynamic range by a factor of $\sim$45.“

  • The statement that “the SiPM be within one standard deviation from both light cones” may be not sufficiently clear. We added a more explicit statement:

“Since light spots exhibit Gaussian intensity profiles, the maximum decrease of light over the SiPM area at the chosen distance is 60\%.”

  • m_i are not moments. We have removed “m_i” from the sentence “for the second and third, the central moments are used.”

  • The units in Eq. 5 were wrong. The formula has been corrected to:
    mu [npe] = \mu [Vs] / (R_L e G)  and “electric” charge is now “electron” charge.

  • 183 “In both cases the mean value of the corrected data is linear” is changed to “In both cases, the linearity of the corrected data is improved.” The discussion about linearity is postponed to the discussion of Fig. 8.

Reviewer 2 Report

Comments and Suggestions for Authors

The paper describes a single-step method for correcting the non-linear response of silicon photomultiplier. For this purpose, the experimental setup, consisting of two pulsed light sources, was built. The procedure for measuring and analyzing the experimental results was shown in detail. The obtained results are no doubt and are clearly presented.

This calibration method can be used for various applications, using a silicon photomultiplier as a detector.

The paper can be published in its current form.

Author Response

Reviewer 2)

We thank the referee for their positive comment.

No change to the text.

Extra corrections.

While reading the paper we have additionally spotted a few more points that needed correction. We excuse ourselves to the reviewers for not spotting this earlier and hope thy will agree with these changes.

  • The definition of dynamic range was not provided, and given the non-linear behaviour it may be misleading. We introduced the concept of “linear dynamic range” with the following definition, provided now in the discussion of Fig. 8. “If the linear dynamic range is defined as the min-to-max range in which the SiPM response is linear within $\pm5\%$, the correction increases the linear dynamic range by a factor of $\sim$45.“

  • The statement that “the SiPM be within one standard deviation from both light cones” may be not sufficiently clear. We added a more explicitly statement:

“Since light spots exhibit Gaussian intensity profiles, the maximum decrease of light over the SiPM area at the chosen distance is 60\%.”

  • m_i are not moments. We have removed “m_i” from the sentence “for the second and third, the central moments are used.”

  • The units in Eq. 5 were wrong. The formula has been corrected to:
    mu [npe] = \mu [Vs] / (R_L e G)  and “electric” charge  is now “electron” charge.

  • 183 “In both cases the mean value of the corrected data is linear” is changed to “In both cases the linearity of the corrected data is improved.” The discussion about linearity is postponed to the discussion of Fig. 8.

Reviewer 3 Report

Comments and Suggestions for Authors

Referee Report: Title: Correcting the Non-linear Response of Silicon Photomultipliers Authors: E. Garutti, J. Schwandt, S. Martens, L. Brinkmann Summary: The manuscript presents a systematic study on correcting the non-linear response of a specific Silicon Photomultiplier (SiPM) model, the Hamamatsu S14160-1315PS. The authors introduce a single-step method to correct the non-linear response of the SiPM, allowing for an expanded dynamic range while maintaining linearity within 5%. The study includes the development of a novel setup for providing homogeneously distributed light from two sources, an LED and a Laser, which can be adjusted in amplitude and arrival time. The correction method is demonstrated to be effective across different operating voltages and integration gate times. Strengths: The study addresses an important issue in SiPM technology and provides a practical method for correcting non-linear responses. The development of a novel setup for light distribution and the event-by-event correction approach adds value to the research. The results show a significant improvement in the dynamic range of the SiPM while maintaining linearity. Suggestions for Improvement: Clarify the specific criteria used to define the "ideal linear response" for the SiPM. Provide more details on the potential limitations or challenges of implementing this correction method in practical applications. Consider discussing the implications of the study findings for other SiPM models or similar photodetectors. Overall, the manuscript presents a well-executed study with valuable insights into correcting the non-linear response of SiPMs. With some minor revisions and additional clarifications, this work has the potential to make a significant contribution to the field of photodetector technology.

Author Response

Suggestions for Improvement:

  1. Clarify the specific criteria used to define the "ideal linear response" for the SiPM.

We thank the referee for the useful suggestions and we try to improve the text accordingly.

We have rephrased the sentence:

“With this method, it is possible to correct the non-linear response of this specific SiPM and expand the dynamic range within 5% of the ideal linear response by a factor of ten. “

to

“With this method, it is possible to correct the non-linear response of this specific SiPM and extend the linear dynamic range by a factor larger than ten.

And we have defined “linear dynamic range”:  “… linear dynamic range is defined as the min-to-max range in which the SiPM response is linear within $\pm5\%$...”

  1. Provide more details on the potential limitations or challenges of implementing this correction method in practical applications.

Thank you for this question. We have added the following sentence in the conclusion.

The main strength of the method presented is that the calibration function can be obtained in a laboratory setup and is independent on the knowledge and linearity of the reference light source.

To apply this in practical applications it will be important to create a test stand where the SiPM is calibrated in conditions as close as possible to the those of the final experiments. Temperature, readout electronics, cooling, light illumination should be close to those expected during operation. But as it is illustrated in this study the correction appears  to be stable against small variations in the operation parameters.  More systematic studies will be needed to quantify this statement.

  1. Consider discussing the implications of the study findings for other SiPM models or similar photodetectors.

The last sentence in the conclusion hints to the need of more systematic studies also using other type of SiPMs. We would like to publish the method first and then enter the more complex comparison and interpretation of results for various SiPM models. Concerning other photodetectors, the method was developed historically from photomultiplier tumes, as mentioned in the reference paper from by Wright.

  1. Overall, the manuscript presents a well-executed study with valuable insights into correcting the non-linear response of SiPMs. With some minor revisions and additional clarifications, this work has the potential to make a significant contribution to the field of photodetector technology.

We thank the referee for their positive comment.

Extra corrections.

While reading the paper we have additionally spotted a few more points that needed correction. We excuse ourselves to the reviewers for not spotting this earlier and hope thy will agree with these changes.

  • The definition of dynamic range was not provided, and given the non-linear behaviour it may be misleading. We introduced the concept of “linear dynamic range” with the following definition, provided now in the discussion of Fig. 8. “If the linear dynamic range is defined as the min-to-max range in which the SiPM response is linear within $\pm5\%$, the correction increases the linear dynamic range by a factor of $\sim$45.“

  • The statement that “the SiPM be within one standard deviation from both light cones” may be not sufficiently clear. We added a more explicitly statement:

“Since light spots exhibit Gaussian intensity profiles, the maximum decrease of light over the SiPM area at the chosen distance is 60\%.”

  • m_i are not moments. We have removed “m_i” from the sentence “for the second and third, the central moments are used.”

  • The units in Eq. 5 were wrong. The formula has been corrected to:
    mu [npe] = \mu [Vs] / (R_L e G)  and “electric” charge  is now “electron” charge.

  • 183 “In both cases the mean value of the corrected data is linear” is changed to “In both cases the linearity of the corrected data is improved.” The discussion about linearity is postponed to the discussion of Fig. 8.